# Complex Presentation of Hao-Fountain Syndrome Solved by Exome Sequencing Highlighting Co-Occurring Genomic Variants

**DOI:** 10.3390/genes13050889

**Published:** 2022-05-16

**Authors:** Manuela Priolo, Cecilia Mancini, Simone Pizzi, Luigi Chiriatti, Francesca Clementina Radio, Viviana Cordeddu, Letizia Pintomalli, Corrado Mammì, Bruno Dallapiccola, Marco Tartaglia

**Affiliations:** 1Unità di Genetica Medica, Grande Ospedale Metropolitano “Bianchi-Melacrino-Morelli”, 89124 Reggio Calabria, Italy; luigichiriatti@libero.it (L.C.); letizia.pintomalli@libero.it (L.P.); corradomammi@tiscali.it (C.M.); 2Area di Ricerca Genetica e Malattie Rare, Ospedale Pediatrico Bambino Gesù, IRCCS, 00146 Rome, Italy; cecilia.mancini@opbg.net (C.M.); simone.pizzi@opbg.net (S.P.); fclementina.radio@opbg.net (F.C.R.); bruno.dallapiccola@opbg.net (B.D.); marco.tartaglia@opbg.net (M.T.); 3Dipartimento di Oncologia e Medicina Molecolare, Istituto Superiore di Sanità, 00161 Rome, Italy; viviana.cordeddu@iss.it

**Keywords:** HAFOUS, *USP7*, cystic fibrosis, *CFTR*, *PKD2*, dual molecular diagnosis, WES

## Abstract

Objective: The co-occurrence of pathogenic variants has emerged as a relatively common finding underlying complex phenotypes. Here, we used whole-exome sequencing (WES) to solve an unclassified multisystem clinical presentation. Patients and Methods: A 20-year-old woman affected by moderate intellectual disability (ID), dysmorphic features, hypertrichosis, scoliosis, recurrent bronchitis, and pneumonia with bronchiectasis, colelithiasis, chronic severe constipation, and a family history suggestive of autosomal dominant recurrence of polycystic kidney disease was analyzed by WES to identify the genomic events underlying the condition. Results: Four co-occurring genomic events fully explaining the proband’s clinical features were identified. A de novo truncating *USP7* variant was disclosed as the cause of Hao–Fountain syndrome, a disorder characterized by syndromic ID and distinctive behavior. Compound heterozygosity for a major cystic fibrosis-causing variant and the modulator allele, IVS8-5T, in *CFTR* explained the recurrent upper and lower respiratory way infections, bronchiectasis, cholelithiasis, and chronic constipation. Finally, a truncating *PKD2* variant co-segregating with polycystic kidney disease in the family allowed presymptomatic disease diagnosis. Conclusions: The co-occurring variants in *USP7* and *CFTR* variants explained the multisystem disorder of the patient. The comprehensive dissection of the phenotype and early diagnosis of autosomal dominant polycystic kidney disease allowed us to manage the *CFTR*-related disorder symptoms and monitor renal function and other complications associated with *PKD2* haploinsufficiency, addressing proper care and surveillance.

## 1. Introduction

Co-occurring pathogenic alleles at multiple loci in a patient can challenge both clinical and molecular diagnosis [1,2]. The presence of phenotypes that do not fit into the pattern of major recognized disorders could either suggest a new condition or support the expansion of a known disease or event(s) indicative of comorbidity [2,3,4]. Genomic sequencing has indeed documented that concomitant pathogenic variants involving two or more genes are found in a significant proportion of patients with rare or ultrarare diseases [1,2,3,4].

De novo heterozygous loss-of-function (LoF) variants in *USP7* (MIM 602519), encoding the ubiquitin-specific protease 7 or herpes virus-associated ubiquitin-specific protease, have recently been identified as the molecular cause of a novel syndromic neurodevelopmental disorder, Hao-Fountain syndrome (MIM 616863) [5,6,7]. To date, 24 individuals with this disease have been reported, including 8 with whole or partial gene deletions.

Cystic fibrosis (MIM 219700) is a systemic autosomal recessive disorder due to LoF variants in the cystic fibrosis transmembrane conductance regulator gene (*CFTR*; MIM 602421), causing a defective transport of chloride and bicarbonate through the respiratory, biliary, gastrointestinal and reproductive epithelia, resulting in the secretion of thick mucus [8,9]. Cystic fibrosis is characterized by wide clinical variability in terms of both disease severity and rate of progression due to multiple causes, including the variable functional impact of pathogenic variants, co-occurrence of modifier alleles, environmental factors, and repeated viral or bacterial infection [10,11,12]. The definition of *CFTR*-related disorder (CFTR-RD) has been introduced to define clinical entities associated with *CFTR* dysfunction not fulfilling the diagnostic criteria of cystic fibrosis [12]. Among the clinical entities included into the CFTR-RD spectrum, disseminated bronchiectasis is regarded as one of the most worrying complications during childhood. An increased incidence of pathogenic *CFTR* variants in bronchiectasis patients has been reported [13], including a consistent association with the IVS8-5T allele (c.1210-12T(5), rs180517, VCV000242535) [12,14,15,16]. *CFTR*-related bronchiectasis is typically characterized by the early-age onset of symptoms, diffuse bronchiectasis with predominant upper lobe distribution on CT scan, and early presence of *Staphylococcus aureus* (*S. aureus)*, *Pseudomonas aeruginosa* (*P. aeruginosa)* or *Burkholdera cepacia* (*B. cepacia)* in respiratory cultures. The IVS8-5T allele is the most common hypomorphic *CFTR* allele worldwide [17], being considered a reduced penetrance variant due to its impact on the proper processing of the *CFTR* transcript [17]. It is widely accepted that 5–10% of normal, full-length *CFTR* transcripts should be sufficient to prevent a severe cystic fibrosis form. Nevertheless, a reduced quantity of transcripts (10–30%) is predicted to result in a CFTR-RD phenotype [18].

Autosomal dominant polycystic kidney disease (MIM 173900) affects 1 in 1000 individuals primarily by the development of large, fluid-filled renal cysts. Extrarenal manifestations include hepatic, pancreatic, and splenic cysts [19,20]. The disorder is caused by LoF variants in *PKD1* (MIM 61313) or *PKD2* (MIM 173910) genes, encoding polycystin-1 and polycystin-2, two proteins that form a complex regulating multiple signaling pathways to maintain normal renal tubular structure and function. The disease is a major cause of end-stage renal disease requiring hemodialysis, peritoneal dialysis, and kidney transplantation [21].

The relatively high incidence of both CFTR-RD and autosomal dominant polycystic kidney disease may justify their occurrence as concomitant events in patients affected by rare diseases, frequently acting as confounding elements and resulting in a more complex phenotype that may delay or mislead the correct main diagnosis. As a paradigmatic example of the importance of reverse phenotyping to properly classify complex diseases resulting from multiple molecular diagnoses, we describe a 20-year-old woman affected with a syndromic intellectual disability disorder characterized by facial dysmorphism, scoliosis, peculiar behavior, short hands and feet associated with bronchiectasis, recurrence of pneumonia, *P. aeruginosa* infections, cholelithiasis and severe chronic constipation, who had a de novo pathogenic variant in *USP7* and was compound heterozygous for pathogenic *CFTR* alleles. She was also found positive for a nonsense variant in *PKD2*, predicting an evolving phenotype including renal and extrarenal manifestations occurring in autosomal dominant polycystic kidney disease. We dissected the contribution of individual variants to the complex clinical manifestations of the patient and characterized the clinical features of the *USP7*-related disease by providing a detailed natural history of the disorder.

## 2. Methods

The proband was evaluated in the frame of the Ospedale Pediatrico Bambino Gesù research program dedicated to undiagnosed patients. The study was approved by the Institutional Ethics Committee (1702_OPBG_2018) and was conducted in accordance with the principles of the Helsinki declaration, after a signed written informed consent was secured by the proband’s parents, who also provided written consent for the publication of the clinical pictures of the patient. Genomic DNA was extracted from circulating leukocytes. Target regions were captured using the Sure Select All Exon v.7 enrichment kit (Agilent, Santa Clara, CA, USA) and sequenced on a Novaseq6000 platform (Illumina, San Diego, CA, USA). WES raw data were processed and analyzed using a previously described in-house implemented pipeline [22,23,24] mainly based on the Genome Analysis Toolkit (GATK) Best Practices [25]. Briefly, the University of California Santa Cruz (UCSC) Genome Reference Consortium Human genome, build 37 (GRCh37)/Human Genome, version 19 (hg19) genome assembly was used as reference for reads alignment by means of the Burrows-Wheeler Aligner-Maximal Exact Match (BWA-MEM) tool [26], and variant calling performed with HaplotypeCaller (GATK v3.7) [25]. Single Nucleotide Polymorphism Effect (SnpEff) v.4.3 [27] and Database for nonsynonymous SNPs’ functional predictions (dbNSFP) v.4.0 [28] tools were used for variants annotation as well as for the in silico prediction of their functional impact by means of Combined Annotation Dependent Depletion (CADD) v.1.6 [29], Mendelian Clinically Applicable Pathogenicity (M-CAP) v.1.3 [30], and Intervar v.2.0.1 [31]. High-quality variants (QUAL > 100, GATK hard-filtering) with frequency <0.1% (gnomAD) and <1% (in-house repository including >2500 exomes) were retained. The assessment of the predicted functional impact was made, assigning a higher priority to variants with CADD score >20, M-CAP score >0.025, tagged as pathogenic/likely-pathogenic by InterVar, as well as taking into account Phen2Gene [32], a tool judging gene-phenotype associations.

## 3. Results

### 3.1. Case Presentation

The proband was a 20-year-old woman born at term from non-consanguineous parents after a pregnancy complicated by threatening abortion. No exposure to teratogens during pregnancy was reported. Her birth weight was 2,500 g (4th centile, −1.76 SD), length 48 cm (26th centile, −0.65 SD), and her head circumference was 33 cm (9th centile, −1.22 SD). APGAR scores were 7 and 9 at 1 and 5 min, respectively. Developmental milestones were delayed. She walked alone at 20 months, her first words were pronounced at 24 months, and language was limited to few words at 36 months. She presented with dysmorphic features including mild synophrys, long and arched eyebrows that were low-set with respect to the upper eyelid, long lashes, generalized hypertrichosis, a bulbous nasal tip, a wide mouth, brachycephaly, bilaterally supernumerary nipples, and short hands and feet (Figure 1). A brain MRI scan revealed the presence of slightly enlarged lateral ventricles with a dysmorphic aspect on the left. Since late childhood, she presented with a peculiar behavior characterized by stubbornness, compulsivity, anxious composure and repeated panic attacks, emotional outbursts and hardly controlled fits of rage, and short attention span requiring psychotherapy with limited benefice. Soon after birth, she suffered from recurrent bronchitis and pneumonia, with confirmed *P. aeruginosa* colonization. Mucus was reported to be thick, and a pulmonary CT scan at 10 years revealed multiple bronchiectasis. She required periodic mechanical mucus aspiration. Gastroesophageal reflux was also reported at birth. The patient showed dysphagia for solid food since the age of 6 months and was fed with a semiliquid diet until 8 years. She also suffered from severe chronic constipation from 6 months of life with the formation of multiple fecaloma requiring periodic manual evacuations. Rectal mucosal biopsy was negative for Hirschprung disease and intestinal neuronal dysplasia. She also experienced biliary sludge and cholelithiasis, which were pharmacologically treated. At the age of 10 years, she was diagnosed with left convex lumbar scoliosis, treated with a corrective corset until 14 years. She also had hypermetropia.

At age of 20 years, her weight was 59 kg (53rd centile, 0.11 SD), height 159 cm (25th centile, −0.65 SD), and head circumference 54 cm (39th centile, −0.28 SD). She presented with ID, mitralic prolapse with sporadic paroxistic tachycardic arrhythmia, polymenorrhea, bilateral supernumerary nipples, and moderate bilateral neurosensory hypoacusia. She walked with a wide based gait and had a peculiar hand posture that tended to be slightly contracted during either walk and standing. She presented with dysphagia for solid food. The objective evaluation confirmed hypertrichosis, synophrys with peculiar eyebrows, deep-set eyes with long palpebral fissures and long lashes, prominent nasal septum extending below the alae nasi, bulbous nasal tip, short and prominent philtrum, and wide mouth with a thin upper lip. She also had low posterior hairline and mildly protruding ears. Renal and hepatic ultrasound assessment excluded the presence of cysts. Serum creatinine and electrolytes were within normal range. CGH-array analysis and *NIPBL*, *SMC1*, *SCM3* and *RAD21* genes testing provided negative results.

The occurrence since early childhood of multiple bronchiectasis, recurrent bronchitis and pneumonia, thick mucus with *P. aeruginosa* colonization, biliary sludge, and cholelithiasis was suggestive of CFTR-RD.

Family history revealed that the proband’s 51-year-old mother presented with renal and hepatic cysts at the age of 30 years. She is currently under surveillance for renal functionality with a slightly increased value for serum creatinine and electrolytes. She also experienced sudden unilateral hearing loss at age of 48 years due to a thrombotic event. A brain MRI showed multiple small gliotic areas compatible with subtle ischemic episodes. Her twin brother was also affected with renal and hepatic cysts and had repeated increased serum creatinine levels. A younger brother (47 years) was affected with renal and hepatic cysts with a normal serum creatinine level, and the 26-year-old sister was affected also with multiple renal cysts. Similarly, the proband’s maternal grandmother (74 years) complained of renal and hepatic cists since the age of 45. She progressed to end-stage renal disease at age of 55 years with increased creatinine levels, and since the age of 71 years, was treated with hemodialysis. All affected members presented with essential hypertension pharmacologically treated with a combined therapy of β-blockers and angiotensin antagonists.

### 3.2. Genetic Analysis Co-Occurring Pathogenic Variants in USP7, CFTR and PKD2 Genes

Singleton-based WES analysis revealed a truncating *USP7* variant, c.1639G>T (p.Glu547*), as a pathogenic event contributing to the phenotype in the proband (Appendix A). LoF variants in this gene underlie Hao–Fountain syndrome. Sanger sequencing on genomic DNA of the proband and both parents validated the variant and confirmed its de novo origin. Whereas the peculiar facial features, intellectual disability, behavioral anomalies, scoliosis, gastroesophageal reflux, short hands and feet, wide based gait, hypotonia, hands contractures during the march and standing are all features of the phenotypic spectrum of this syndromic disorder (Table 1), the recurrent pulmonary infections, thick mucus, bronchiectasis and *P. aeruginosa* pneumonia, biliary sludge and cholelithiasis are not causally related to *USP7* haploinsufficiency. WES data reanalysis was carried out to identify the molecular cause of the CFTR-RD features revealing a pathogenic variant in *CFTR*, c.1521_1523delCTT (Phe508Del, rs113993960, VCV000634837), the gene mutated in cystic fibrosis. Sanger sequencing of parental DNA documented that the variant was inherited from the clinically unaffected mother. Based on this finding, a manual re-evaluation of WES data allowed us to identify a second variant, c.1210-12T(5) (IVS8-5T, rs1805177), representing a pathogenic modulator allele in cystic fibrosis (VCV000242535), which was paternally inherited, thus confirming a diagnosis of CFTR-RD. Of note, the proband was also found to be heterozygous for a previously reported pathogenic variant in *PKD2* (c.295G>T, p.Glu99*; VCV000827765) without manifesting disease symptom. Segregation analysis was then performed on the available family members who had features fitting with autosomal dominant polycystic kidney disease and documented the pathogenic *PKD2* variant in each subject. Less stringent filtering criteria (MAF <5%, considering both gnomAD and in-house databases) excluded the presence of “pathogenic” or “likely pathogenic” variants in genes known to be implicated in sensorineural hearing loss.

### 3.3. Facial Features Assessment in Hao-Fountain Syndrome

We reviewed the available clinical data on Hao–Fountain syndrome to assess the consistency of dysmorphic features and clinical signs (Table 2). Although previous reports failed in identifying a recognizable facial gestalt [5,6,7], recurrent features were appreciated, including a prominent nasal septum extending below the alae nasi and deep-set eyes. A blind evaluation of available clinical pictures by three experienced clinical geneticists (M.P., F.C.R., B.D.) disclosed other recognizable signs, including long eyebrows that were low-set with respect to the upper eyelid (12/15, 80%), long palpebral fissures (14/16, 87%), a short and prominent philtrum (13/16, 81% and 11/16, 68%, respectively), and a thin upper lip (11/16, 68%) (Table 2). This confirmed the previous assessment, in which almost all subjects presented with a prominent nasal septum (13/15, 87%) extending below the alae nasi (16/16) and deep-set eyes (15/15). The facial gestalt of the disorder, as described above, was well-recapitulated by our proband (Figure 1).

## 4. Discussion

Hao–Fountain syndrome is caused by de novo variants in the *USP7* gene [5,6,7], which encodes a deubiquitinating enzyme of the ubiquitin-specific protease family, promoting the cleavage of multiple chain linkages [5,6,33,34,35]. USP7 regulates the ubiquitination status of many proteins, including multiple tiers of the MDM2-p53 pathway, a cascade controlling various physiological processes, such as DNA repair, cell cycle checkpoints, transcription, immune responses, and viral replication suppression [33,34]. The spectrum of the reported pathogenic *USP7* variants (eight missense, five truncating, and three splice-site variants), as well as heterozygous deletions involving *USP7* in eight individuals, support haploinsufficiency as the mechanism of disease. 

The main Hao–Fountain syndrome clinical features and signs are summarized in Table 1 and Table 2. By reviewing the previous series and comparing them with the present case, we confirmed the presence of a recognizable disorder characterized by developmental delay/intellectual disability (24/25, 96%), speech delay (25/25, 100%), dysmorphic face (20/22, 91%), MRI anomalies (13/17, 76%), hypotonia (16/22, 72%), behavioral anomalies (15/22, 68%), gastroesophageal reflux with feeding difficulties (11/18, 61% and 13/23, 56%, respectively) and eye abnormalities/vision impairment (16/25, 64%). Additional problems, ranging from aggressive behavior, fits of rage, impulsivity, compulsivity, stubbornness, manipulative approach, anxious component, and panic attacks, depict a distinct recurring behavioral phenotype to be considered in the genetic counseling of individuals with Hao–Fountain syndrome. In the present case, the behavioral component was a major concerning problem because of the scarcity of patient compliance to therapies and pharmacological treatments, which commonly triggered temper tantrums. In Hao–Fountain syndrome, autism spectrum disorder (9/17, 53%) and attention deficit-hyperactivity disorder (7/17, 41%) can also occur but were not documented in the present patient. Seizures/EEG abnormalities, neonatal hypotonia, and brain MRI anomalies are also common (Table 1). Ventricular enlargement and neonatal hypotonia were observed in our patient. An abnormal wide-based gait occurs in 47% of cases (7/15), while short stature and scoliosis/kyphosis has been recorded in one-third of individuals (6/21 and 7/21, respectively). Small hands and feet were recorded in approximately 25% of cases (Table 1 and Figure 1). Our patient also had neurosensory hypoacusia, in line with hearing difficulties reported in two other cases. We also observed generalized hypertrichosis involving the face, back, arms and legs, so far unreported in Hao–Fountain syndrome.

Based on published data and the present patient, we identified recurrent distinctive facial features, including low-set eyebrows, deep-set eyes with long palpebral fissures, prominent nasal septum extending below the alae nasi, short and prominent philtrum, and thin upper lip (Table 2). Previous reports failed to recognize these features possibly because of ascertainment bias, based on the comparison of individuals at different ages. A detailed iconographic and clinical review of our patient in her first 20 years of life allowed us to compare her features with those of other individuals with Hao–Fountain syndrome at different ages (see iconographic material in [6] and Figure 1).

Our patient presented with severe chronic constipation resistant to treatment, likely due to the concomitant CFTR-RD, which occasionally has been reported [36]. In our patient, the presence of biliary sludge and cholelithiasis, and bronchiectasis with repeated *P. aeruginosa* colonization are also undoubtedly related to the CFTR-RD, which prompted the inclusion of the patient into a dedicated program of surveillance and follow-up.

The proband is also heterozygous for a nonsense *PKD2* variant segregating in several maternal autosomal dominant polycystic kidney disease-affected members. Symptomatic patients presented with renal/hepatic cysts and progressive renal insufficiency. At present, our patient does not show renal/hepatic involvement and has normal renal functionality. She is currently under strict follow-up with periodic evaluations. She is also monitored to prevent hypertension and vascular accidents, being affected with mitralic prolapse and paroxistic tachycardic arrhythmia. Consistently, patients with autosomal dominant polycystic kidney disease have an increased rate of cardiovascular abnormalities, including hypertension, left ventricular hypertrophy, aortic root dilatation, arterial aneurysms, heart valve abnormalities, and intracranial aneurysms [10,37]. The vascular involvement is directly related to the reduced expression of the polycystin protein in the endothelial cells and vascular smooth muscle cells of blood vessels [10,38]. Indeed, all the maternal affected members presented with variable combinations of essential hypertension and cardiovascular accidents.

In conclusion, we describe a patient affected by a complex clinical phenotype resolved by WES carrying multiple molecular pathogenic variants, including a de novo inactivating *USP7* variant, compound heterozygosity for a major cystic fibrosis-causing variant and a modifier allele, IVS8-5T, determining a CFTR-RD condition, and a maternally inherited truncating *PKD2* variant predisposing to adult-onset autosomal dominant polycystic kidney disease. We also document the occurrence of a distinctive facial gestalt in Hao–Fountain syndrome by defining the recurrent facial features. The present finding further confirms the utility of WES in solving complex phenotypes and the need for the accurate analysis and interpretation of the genomic data as a key to properly managing patients. From a clinical perspective, recognizing multiple molecular causes has several implications for counseling, treatment, and proper follow-up and long-term surveillance.

## Figures and Tables

**Figure 1 genes-13-00889-f001:**
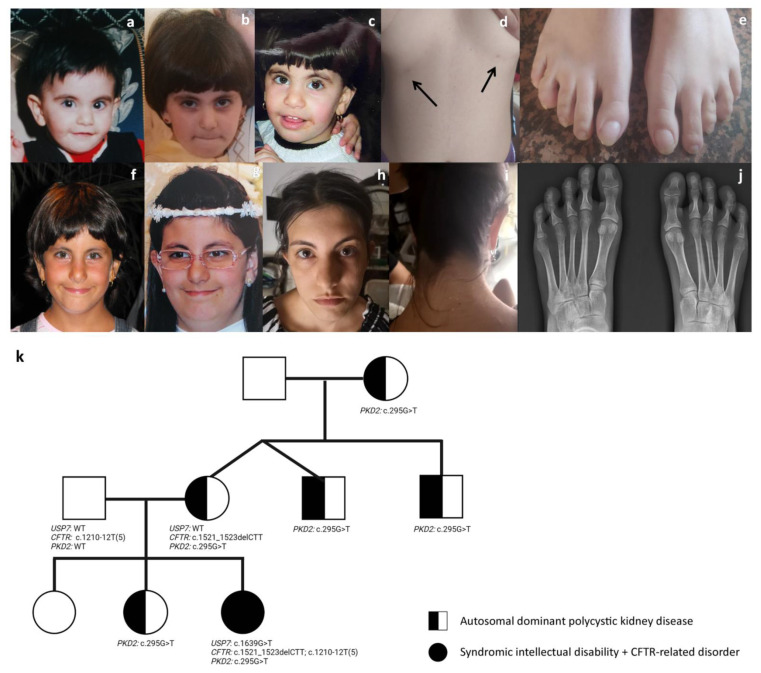
Clinical features of the proband at different ages, and family pedigree reporting phenotypes and genotypes of individual members. Facial features at 1 year (**a**), 3 years (**b**), 4 years (**c**), 7 years (**f**), 10 years (**g**), and 20 years (**h**). The proband shows bilateral supernumerary nipples (black arrows) (**d**), short feet (**e**,**j**), and low posterior hairline (**i**). The tree of the family shows transmission of the pathogenic *PKD2* and *CFTR* variants, and the de novo occurrence of the pathogenic *USP7* variant (**k**).

**Table 1 genes-13-00889-t001:** Clinical features of patients with pathogenic *USP7* variants.

Reported Cases	N = 8	N = 5	N = 8	N = 3	Present Report	Total, N = 25
*USP7 variant*	gene deletions ^a,b^	truncating variants ^b,c^	missense variants ^b^	splice site variants ^b^	truncating variant	
Sex	5M, 3F	1M, 4F	2M, 6F	3M	F	11M, 14F
Dysmorphic facial features	4/6	5/5	7/7	3/3	+	20/22
**DEVELOPMENT**						
Developmental delay/intellectual disability	8/8	5/5	7/8	3/3	+	24/25
Decreased fetal movement	0/6	1/3	1/3	0/3	+	3/16
Neonatal hypotonia	1/6	5/5	4/7	0/3	+	11/22
Hypotonia	4/7	5/5	5/7	1/3	+	16/22
Speech delay	8/8	5/5	8/8	3/3	+	25/25
Nonverbal	1/8	0/3	3/8	0/3	-	4/22
Walking milestone age (months)	28 (mean)	25 (mean)	57 (mean)	21 (mean)	20	
Sitting without support (months)	14 (mean)	11.6 (mean)	17 (mean)	11 (mean)	12	
**NEUROLOGICAL**						
Abnormal MRI	3/4	2/4	7/7	0/1	+	13/17
Seizures	4/8	3/5	3/7	0/3	-	10/24
Abnormal gait	2/5	0/2	4/4	0/3	+	7/15
**BEHAVIOR**						
Behavioral anomalies	7/8	1/3	2/7	2/3	+	15/22
Autism spectrum disorder	7/7	0/3	2/4	0/3	-	9/17
attention deficit-hyperactivity disorder	3/7	0/2	1/4	2/3	+	7/17
Skin picking	2/8	0/4	1/5	0/3	-	3/21
**GASTROINTESTINAL**						
Feeding problems, need for special feeding tools	4/7	2/4	4/8	2/3	+	13/23
Gastroesophageal reflux	3/5	2/3	4/6	1/3	+	11/18
Difficulty in gaining weight	1/6	1/3	5/7	2/3	+	10/20
Chronic constipation	3/5	1/2	1/6	0/3	+	6/17
Neonatal poor suck	2/6	2/5	1/4	0/3	+	6/19
Excessive weight gain	0/6	0/4	0/6	3/3	-	3/20
**RESPIRATORY**						
Asthma	1/4	2/3	1/4	2/3	-	6/15
Sleep apnea/sleep disturbance	3/7	1/3	1/7	0/3	+	6/21
**SKELETAL**						
Short stature	2/7	1/4	3/6	0/3	-	6/21
Scoliosis/kyphosis	2/6	0/4	1/7	3/3	+	7/21
Contractures	2/6	2/4	0/4	0/3	+	5/18
Small hands	2/6	2/4	0/5	0/3	+	5/19
Small feet	1/6	2/4	1/6	0/3	+	5/20
Hip dysplasia	0/6	0/4	2/8	0/3	+	3/22
**SENSORY SYSTEM**						
Eye abnormalities	6/8	4/5	4/8	1/3	+	16/25
Hearing difficulties	1/7	1/4	0/8	0/3	+	3/23

Modified from Ref. [6]. M, male; F, female. ^a^ Ref. [5]. ^b^ Ref. [6]. ^c^ Ref. [7].

**Table 2 genes-13-00889-t002:** Facial features of individuals with pathogenic *USP7* variants.

Dysmorphic Features (Human Phenotype Ontology Term)	Fountain et al., 2019 [6]	Capra et al., 2020 [7]	Present Case	Total
Low-set eyebrows with respect to the upper eyelid (NA)	(11/14)	NA	+	12/15(80%)
Deeply set eyes (HP:0000490)	14/14	NA	+	15/15 (100%)
Prominent nasal septum (HP:0005322)	12/14	NA	+	13/15 (87%)
Low hanging septum of nose (HP:0005322)	14/14	+	+	16/16 (100%)
Long palpebral fissures (HP:0000637)	12/14	+	+	14/16 (87%)
Long eyelashes (HP:0000527)	NA	NA	+	
Protruding ears (HP:0000411)	NA	NA	+	
Short philtrum (HP:0000322)	12/14	-	+	13/16 81%
Prominent philtrum (HP:0002002)	10/14	-	+	11/16 68%
Thin upper lip (HP:0000219)	9/14	+	+	11/16 68%
Low posterior hairline (HP:0002162)	NA	NA	+	

NA, not available.

## Data Availability

The WES data supporting the findings of this study are available on request from the corresponding author (M.P.). The data are not publicly available due to privacy/ethical restrictions.

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
