# Peer review of "Complex Presentation of Hao-Fountain Syndrome Solved by Exome Sequencing Highlighting Co-Occurring Genomic Variants"

_genes, 2022, doi:10.3390/genes13050889_

Round 1
Reviewer 1 Report
Priolo and colleagues submit a case report of a patient presenting with symptoms of different disease entities whereby the complex phenotype could be (mostly) resolved upon whole exome sequencing. They demonstrate that co-occurrence of variants causative for HAO-Fountain syndrome, for CFTR-related disorders or atypical cystic fibrosis and for autosomal dominant polycystic kidney disease explain most aspects of the phenotype observed.
The clinical data presented is very detailed and the authors' dissection of which genetic variant is likely responsible for which symptoms is highly credible. In fact, the entire case report would be excellently suited for the MDPI "Journal of Personalized Medicine" where it might be appreciated by the audience who seeks to understand their patient's complex phenotype and adequately treat their disease(s).
The reviewer can make only few suggestions to improve the present manuscript, however, the case report would benefit from the following additional work on the manuscript:
- by its very nature, this case report is rich in abbreviations. These capital-letter-abbreviations are given for genes (reasonable and unavoidable), for tools (GATK) and for diseases (HAFOUS). However, as the abbreviated vocabulary of "tools" and "disease entities" are not shared by all members of the audience, a sentences meaning can easily be obscured, though maybe HAFOUS can be an exception to this proposal as we can all learn one or two acronyms used frequently in this manuscript. The reviewer recommends to restrict abbreviations to genes in order to enable an easy access to the manuscript's content. For the next few examples listed, please keep in mind that not all readers will be familiar with the fact that there is a human phenotype ontology - and some will be more familiar with SNOMED CT as are more affiliated with healthcare, not with research. Likewise, ADPKD might be mentally linked to an ADP-converting enzyme by readers who are used to gene lists but do not practice medicine. While italics for genes are used precisely because of that, not all readers are aware of that rule and its reason. Thus: Please avoid unnecessary abbreviations, even if it is technically allowed to do so after defining it once in brackets, and whenever possible, spell out and/or explain (“the disease ADPKD” instead of “ADPKD”):
- line 74 – ESRD
- throughout the manuscript HAFOUS, ADPKD – please check whether at the beginning of a new chapter in the manuscript, a rephrasing like “the disease HAFOUS” might remind the reader what HAFOUS was standing for
- The methods paragraph on WES would benefit from being edited for easier readability. This might be accomplished by specifying whether the acronym denotes a tool or a threshold for a parameter (and what that parameter stands for).
- Table 2, heading of column 1: HPO, please spell out and clarify precisely (the HP-number is not equivalent of the ontology, but of an entry within the ontology)
- Line 233: ASD, ADHS: please spell out
- Introduction, line 48: CFTR is a chloride and bicarbonate transporter – to avoid that the old annotation persists, please add the bicarbonate. Also, the references for CFTR are partially outdated. For instance, reference 18 from 2005 is an editorial – it does not correspond to the original finding of “less CFTR transcript means an increased risk of CFTR-RD”. A timely review that explains CF and CFTR in most of its aspects would be https://pubmed.ncbi.nlm.nih.gov/31570318/; to ensure transparency: the reviewer is not an author of this review.
- Line 94: consent; please specify whether consent was given by the patient or by the legal guardian; in relation to the portraits shown in Figure 1: how was permission obtained?
- Methods / WES. Apparently, the authors choose to select established algorithms for processing the primary data, for detecting variants and for annotation of the detected variants. Do the guidelines for selection of these algorithms specify what can be missed? In other words, is there an estimate how likely it is to overlook a variant, given the specifics of the data set analysed here (coverage, quality of primary reads)? For clarification: this is not to discredit the data, but to specify “WES raw data were processed” rather in metadata commonly used to describe next-generation sequencing data. Also, the discussion (laudable approach!) states that WES was done from parents and offspring – hence, basic auality data of all three next generation nseqeuncing data sets should be provided here. This will enable fellow researchers to judge wherther their own WES pipeline would be comparable in terms of primary reads, and it will also ensure that the data set can be judged in relation to future next-generation sequencing techniques.
- Line 136 – sentence “She was also had hypermetropia.” Typo / grammatically incorrect.
- Line 257: “Our patient does not manifest …” – likely: “our patient does not show or”, or “in our patient, XYZ does not manifest ..”
- Concluding paragraph, line 268 ff. Certainly, there are limitations to this method – in other words: are there still symptoms and disease manifestations that are not explained by the variants uncovered by WES in this patient? If yes, these whould be summarized as well in the concluding remarks. Otherwise, “fully resolved by WES” or “mostly resolved by WES” would be a good qualification.
Author Response
COMMENTS FROM REVIEWER #1
We wish to thank the reviewer for her/his positive evaluation and the constructive remarks.
COMMENT (1):
“… by its very nature, this case report is rich in abbreviations. These capital-letter-abbreviations are given for genes (reasonable and unavoidable), for tools (GATK) and for diseases (HAFOUS). However, as the abbreviated vocabulary of "tools" and "disease entities" are not shared by all members of the audience, a sentences meaning can easily be obscured, though maybe HAFOUS can be an exception to this proposal as we can all learn one or two acronyms used frequently in this manuscript. The reviewer recommends to restrict abbreviations to genes in order to enable an easy access to the manuscript's content.
Table 2, heading of column 1: HPO, please spell out and clarify precisely (the HP-number is not equivalent of the ontology, but of an entry within the ontology).
Line 233: ASD, ADHS: please spell out.
Introduction, line 48: CFTR is a chloride and bicarbonate transporter – to avoid that the old annotation persists, please add the bicarbonate.”
AUTHORS’ REPLY: Following the reviewer’s advice, we revised the text to remove unnecessary abbreviations. We also revised the paragraph dealing with the methods (WES data analysis) and both Tables 1 and 2, accordingly.
COMMENT (2):
“… the references for CFTR are partially outdated. For instance, reference 18 from 2005 is an editorial – it does not correspond to the original finding of “less CFTR transcript means an increased risk of CFTR-RD”. A timely review that explains CF and CFTR in most of its aspects would be https://pubmed.ncbi.nlm.nih.gov/31570318/; to ensure transparency: the reviewer is not an author of this review.”
AUTHORS’ REPLY: We revised the references following the reviewer’s remark.
COMMENT (3):
“Line 94: consent; please specify whether consent was given by the patient or by the legal guardian; in relation to the portraits shown in Figure 1: how was permission obtained?”
AUTHORS’ REPLY: We thank the reviewer for catching the missing information. We confirm that a signed informed consent for the publication of the patient’s photographs was obtained. We revised the first paragraph of the “Methods” section accordingly.
COMMENT (4):
“Methods / WES. Apparently, the authors choose to select established algorithms for processing the primary data, for detecting variants and for annotation of the detected variants. Do the guidelines for selection of these algorithms specify what can be missed? In other words, is there an estimate how likely it is to overlook a variant, given the specifics of the data set analysed here (coverage, quality of primary reads)? For clarification: this is not to discredit the data, but to specify “WES raw data were processed” rather in metadata commonly used to describe next-generation sequencing data.”
AUTHORS’ REPLY: As reported in the “Methods” section, the WES data analysis workflow follows the GATK Best Practices (Van der Auwera et al., 2013, Curr Protoc Bioinform, 43), which are recognized as the gold standard. Moreover, in line with the reviewer’s note, Supplemental Table 1 reports the WES data statistics and output to provide information on the quality of the obtained sequencing data.
COMMENT (5):
“Also, the discussion (laudable approach!) states that WES was done from parents and offspring – hence, basic quality data of all three next generation nseqeuncing data sets should be provided here. This will enable fellow researchers to judge wherther their own WES pipeline would be comparable in terms of primary reads, and it will also ensure that the data set can be judged in relation to future next-generation sequencing techniques.”
AUTHORS’ REPLY: We thank the reviewer for catching this inaccuracy, which was amended in the revised manuscript (Abstract, Results and Discussion). As correctly reported in Supplemental Table 1, WES was performed in the proband only, while variant validation and segregation analyses were extended to the parents (USP7 and CFTR) or to other members of the family (PKD2).
COMMENTS (6) and (7):
“Line 136 – sentence “She was also had hypermetropia.” Typo / grammatically incorrect.”
“Line 257: “Our patient does not manifest …” – likely: “our patient does not show or”, or “in our patient, XYZ does not manifest ..””
AUTHORS’ REPLY: The text was revised accordingly.
COMMENT (8):
“Concluding paragraph, line 268 ff. Certainly, there are limitations to this method – in other words: are there still symptoms and disease manifestations that are not explained by the variants uncovered by WES in this patient? If yes, these whould be summarized as well in the concluding remarks. Otherwise, “fully resolved by WES” or “mostly resolved by WES” would be a good qualification.”
AUTHORS’ REPLY: The text was revised following the reviewer’s advice.

Reviewer 2 Report
The manuscript by Priolo et al. describes the identification of 4 different pathogenic genetic variants in 3 different genes underlying a complex phenotype in a 20-year old female. This phenotype results from the superposition of up to three different disorders: Hao-Fountain syndrome (caused by a de novo nonsense variant p.Glu547 in the USP7 gene), CFTR-related disorder (caused by compund heterozygosity for the Phe508del and hypomorph IVS8-5T at the CFTR gene) and as-yet-asymptomatic, autosomal dominant polycystic kidney disease (caused by a nonsense variant in the PKD2 gene, p.Glu99, which segregated in the family, ).
The paper is informative and very clearly written; conclusions are sound. Thus I only have the following minor comments:
(1) Discussion, line 239: "Our patient also had neurosensory hypoacusia, in line with hearing difficulties reported in two other cases". Sensorineural hearing loss (SNHL) is the most common presentation of hereditary hypoacusia, which may be caused by mutations in many genes (so far about 100 genes identified and up to 250 deemed to be involved), with an overall frequency of 1/1000 to 1/2000 births. The low frequency of hearing loss among clinical manifestations in Hao-Fountain syndrome (just 3/23 reported cases) amkes me wonder whether the SNHL observed in the patient is truly the consequence of the UPS7 mutation or rather due to a mutation in one of the known or unknown HL genes. Given that the authors specifically searched for very low-frequency variants in the general population (gnomAD MAF<0.1%), some relatively common variants in known genes causing SNHL may have escaped their attention (e.g. GJB2 c.35delG, with a carrier rate in Italy of 1/40 or 2.5%). Hence, I believe that the authors should not exclude that maybe the SNHL observed in their patient is not due to the USP7 de novo variant, but instead might be due to putatively unidentified pathogenic variants in any of the genes underlying SNHL.
(2) Line 56: "complicacy" should read "complication".
(3) Line 136: "was" should be deleted.
(4) For the sake of clarity regarding the segregation in the family of the variants in CFTR and PKD2, I suggest including the family pedigree, now presented as a supplementary figure; in one of the figures of the main text.
Author Response
COMMENTS FROM REVIEWER #2
We thank the reviewer for the positive assessment of our work.
MINOR COMMENT (1):
“Discussion, line 239: "Our patient also had neurosensory hypoacusia, in line with hearing difficulties reported in two other cases". Sensorineural hearing loss (SNHL) is the most common presentation of hereditary hypoacusia, which may be caused by mutations in many genes (so far about 100 genes identified and up to 250 deemed to be involved), with an overall frequency of 1/1000 to 1/2000 births. The low frequency of hearing loss among clinical manifestations in Hao-Fountain syndrome (just 3/23 reported cases) makes me wonder whether the SNHL observed in the patient is truly the consequence of the UPS7 mutation or rather due to a mutation in one of the known or unknown HL genes. Given that the authors specifically searched for very low-frequency variants in the general population (gnomAD MAF<0.1%), some relatively common variants in known genes causing SNHL may have escaped their attention (e.g. GJB2 c.35delG, with a carrier rate in Italy of 1/40 or 2.5%). Hence, I believe that the authors should not exclude that maybe the SNHL observed in their patient is not due to the USP7 de novo variant, but instead might be due to putatively unidentified pathogenic variants in any of the genes underlying SNHL.”
AUTHORS’ REPLY: The reviewer raised a fair issue. Following her/his constructive remark, we assessed all annotated variants with gnomAD MAF<5% and having a frequency <5% in our population-matched in-house database (>2500 exomes). In particular we checked for variants involving SNHL-related genes, including GJB2. We did not find functionally/clinically relevant variants that could explain SNHL, even though we cannot exclude that causative variant(s) might be missed due to the sequencing approach that was limited to the coding sequences.
MINOR COMMENTS (2), (3) and (4):
“Line 56: "complicacy" should read "complication".”
“Line 136: "was" should be deleted.”
“For the sake of clarity regarding the segregation in the family of the variants in CFTR and PKD2, I suggest including the family pedigree, now presented as a supplementary figure; in one of the figures of the main text.”
AUTHORS’ REPLY: We thank the reviewer for catching our inaccuracies, which were amended in the revised version of the manuscript. Following the reviewer’s suggestion, the family tree was included in Figure 1
